# Human Wharton’s Jelly-Derived Mesenchymal Stem Cells Minimally Improve the Growth Kinetics and Cardiomyocyte Differentiation of Aged Murine Cardiac c-kit Cells in In Vitro without Rejuvenating Effect

**DOI:** 10.3390/ijms20225519

**Published:** 2019-11-06

**Authors:** Wai Hoe Ng, Yoke Keong Yong, Rajesh Ramasamy, Siti Hawa Ngalim, Vuanghao Lim, Bakiah Shaharuddin, Jun Jie Tan

**Affiliations:** 1Advanced Medical and Dental Institute, Universiti Sains Malaysia, Bertam 13200, Kepala Batas, Penang, Malaysia; nwh14_ipg017@student.usm.my (W.H.N.); siti.hawa.ngalim@usm.my (S.H.N.); vlim@usm.my (V.L.); bakiah@usm.my (B.S.); 2Department of Human Anatomy, Faculty of Medicine and Health Sciences, Universiti Putra Malaysia, Serdang 43400, Selangor Darul Ehsan, Malaysia; yoke_keong@upm.edu.my; 3Department of Pathology, Faculty of Medicine and Health Sciences, Universiti Putra Malaysia, Serdang 43400, Selangor Darul Ehsan, Malaysia; rajesh@upm.edu.my

**Keywords:** aged cardiac c-kit cells, wharton’s jelly-derived mesenchymal stem cells, co-culture, cardiomyocyte differentiation

## Abstract

Cardiac c-kit cells show promise in regenerating an injured heart. While heart disease commonly affects elderly patients, it is unclear if autologous cardiac c-kit cells are functionally competent and applicable to these patients. This study characterised cardiac c-kit cells (CCs) from aged mice and studied the effects of human Wharton’s Jelly-derived mesenchymal stem cells (MSCs) on the growth kinetics and cardiac differentiation of aged CCs in vitro. CCs were isolated from 4-week- and 18-month-old C57/BL6N mice and were directly co-cultured with MSCs or separated by transwell insert. Clonogenically expanded aged CCs showed comparable telomere length to young CCs. However, these cells showed lower *Gata4*, *Nkx2.5*, and *Sox2* gene expressions, with changes of 2.4, 3767.0, and 4.9 folds, respectively. Direct co-culture of both cells increased aged CC migration, which repopulated 54.6 ± 4.4% of the gap area as compared to aged CCs with MSCs in transwell (42.9 ± 2.6%) and CCs without MSCs (44.7 ± 2.5%). Both direct and transwell co-culture improved proliferation in aged CCs by 15.0% and 16.4%, respectively, as traced using carboxyfluorescein succinimidyl ester (CFSE) for three days. These data suggest that MSCs can improve the growth kinetics of aged CCs. CCs retaining intact telomere are present in old hearts and could be obtained based on their self-renewing capability. Although these aged CCs with reduced growth kinetics are improved by MSCs via cell–cell contact, the effect is minimal.

## 1. Introduction

Cardiovascular disease remains the number one killer disease and is responsible for approximately 30% of all deaths caused by non-communicable diseases worldwide [1]. Although advances in pharmacological and surgical interventions have substantially prolonged the lifespan of myocardial infarcted patients, cardiomyocyte loss and fibrosis following ischemia introduce the beating heart with a non-contractile scar tissue, causing it to remodel and fail if it is left untreated. Whole heart transplantation may cure end-stage heart failure, but it is limited by donor shortage and high risk of host-versus-graft immune rejection [2,3].

Cell therapy has emerged as a promising strategy to attenuate the progression of the disease and prevent heart remodelling. Infusion of autologous cardiac-derived c-kit cells had shown significantly improved the global left ventricular ejection fraction (LVEF) by 10% at 12-month follow up in patients in Stem Cell Infusion in Patients with Ischemic CardiOmyopathy (SCIPIO) trial in 2011 [4,5]. Although autologous cell source is preferred, the cell function could be limited by the effect of ageing. Study have shown that heart c-kit cell number was reduced in old patients and could be further complicated by diabetes and chronic heart disease [6]. Another study also showed that ageing and chronic heart failure can deteriorate functionally competent cardiac c-kit cells [7], leaving the question whether the properties and characteristics of cardiac c-kit cells from aged heart are still similar to that of from the young heart.

Several reports showed that cardiac c-kit cell function can be enhanced in the presence of bone marrow mesenchymal stem cells (MSCs) and therapy with combined human cardiac c-kit cells and bone marrow MSCs significantly reduced infarct size, improved left ventricular function and restored cardiac function in infarcted swine hearts [8]. Here, we sought to isolate cardiac c-kit cells (CCs) from young (one month) and aged (18 months) mice, and examine the changes of growth kinetics and cardiac differentiation in human Wharton’s Jelly derived MSC co-culture.

## 2. Results

### 2.1. Isolation and Characterisation of CCs

The CCs were obtained from the whole heart of 1- and 18-month-old C57/BL6N mice upon mechanical and enzymatic digestion. CCs were enriched based on c-kit expression using magnetic sorting. Due to initial low cell yield following fresh isolation, characterisation of isolated CCs was performed between passage 5 to 8. CCs isolated from 1-month old mice hearts are termed young CCs (yCCs), and the CCs from 18-month-old mice are called aged CCs (aCCs). Characterisation of the expanded primary c-kit cells was performed using flow cytometry (Appendix A). In order to obtain a more homogenous CC population, both populations were subjected to single cell clonogenic expansion. Wells with CC colony of more than 50 cells were selected for expansion. Only the fast-growing cells which formed colonies from single cells within 2–3 weeks were selected for further characterisation using flow cytometry. Most of the clonogenic yCCs expressed c-kit (89.4 ± 2.8%), SCA-1 (60.9 ± 11.7%), CD105 (95.1 ± 2.4%), CD166 (93.3 ± 2.5%), with negligible mesenchymal marker expression CD90 (2.8 ± 0.8%). CD45 represented only 0.5 ± 0.2% of the yCC population (Figure 1A, B). Likewise, majority of aCCs expressed c-kit (93.8 ± 4.4%), SCA-1 (79.4 ± 2.8%), CD105 (98.6 ± 0.8%). CD34+ and CD45+ population only expressed in 5.3 ± 2.0% and 0.2 ± 0.1% of the total aCC population, respectively. However, higher CD90+ (99.6 ± 0.1%) vs. CD140a+ (63.4 ± 15.2%) populations were found in aCCs as compared to yCCs; whereas only 0.3 ± 0.3% expressed CD166^+^ (Figure 1A, C). Collectively, proliferative yCCs were found to be CD90^Neg^CD140^Neg^CD166^Pos^, while aCCs were CD90^Pos^CD140^Pos^CD166^Neg^. Immunocytochemistry analysis showed that both CCs expressed c-kit, cardiac transcription factors (GATA4, NKX2.5), proliferative marker (Ki67), and pluripotent markers (OCT3/4 and SOX2) (Appendix A) but were negative for mast cell marker (Tryptase) (Appendix A), which was in line with other studies [7,9,10,11,12].

### 2.2. Characterisations of CCs Isolated from Young and Aged Mice 

To examine the clonogenicity of yCCs and aCCs, the cells were grown at 1 cell/2 wells for two weeks. As speculated, aCCs showed lower clonogenicity (9.5 ± 2.9%) as compared to yCCs (21.2 ± 4.4%) (*p* < 0.01) (Figure 2A). The yCCs were comparatively more proliferative in average than the aCCs. No statistical difference was observed in the average doubling time for primary yCCs (26.4 ± 1.7-h) and aCCs (27.9 ± 0.8-h) (Figure 2B,C). Clonogenic yCCs showed an average doubling time of 17.2 ± 0.4-h, which was shorter than the clonogenic aCCs (25.0 ± 0.7-h, *p* < 0.001). Although clonal expansion shortened the average doubling time of aCCs to 25.0 ± 0.7-h, it was not statistically different from the primary isolated population (27.9 ± 0.8-h; *p* = 0.292).

In the primary yCCs, about 65% were in G1 and 31% were in G2 phase (Figure 2D). The primary aCCs were having similar percentage of cells at G1 phase (64%) but with almost 2.5-fold lower percentage of cells in G2 phase as compared to primary yCCs. When CCs were clonogenically expanded, the percentage of cells resided in G1 phase increase to 88% for clonogenic yCCs and 71% for clonogenic aCCs. With almost similar percentage of cells resided in S phase, clonogenic aCCs showed 17% more cells resided in G2 phase (8% vs. 25%).

Age-related changes in telomere length and telomerase activity in both CCs were assessed by qPCR. Comparing within primary CCs, aCCs had significantly shorter relative telomere length (0.24 ± 0.03 vs. 0.78 ± 0.19 T/S ratio; *p* < 0.01; Figure 2E) but higher relative telomerase activity (1.10 ± 0.01 vs. 0.94 ± 0.03 unit) as compared to yCCs (*p* < 0.01) (Figure 2F). Surprisingly, clonogenic aCCs showed comparable relative telomere length (0.94 ± 0.02 vs. 1.0 ± 0.07 T/S ratio; *p* = 0.987) and relative telomerase activity (1.09 ± 0.02 vs. 1.01 ± 0.01 unit; *p* = 0.894) to clonogenic yCCs. 

Although clonogenic aCCs exhibited comparable telomere length and telomerase activity as compared to young counterpart, we tested whether the growth kinetics are similar to that of clonogenic yCCs. Population doubling time, cell cycle, migration, cardiostem spheres (CSp) formation, trilineage differentiation, and stemness gene expression were assessed. 

The CSp formation of clonal aCCs were significantly impaired, with only 4 ± 1 spheres as compared to yCCs with 64 ± 19 spheres per 20 000 cells (*p* = 0.013) (Figure 2G). Upon harvesting of CSps, the spheres were evaluated for their ability to perform trilineage differentiation spontaneously in leukaemia inhibitory factor- (LIF-) deprived CCs medium. Clonogenic yCCs were able to differentiate into smooth muscle and endothelial lineages but did not spontaneously form cardiomyocytes. Nevertheless, aCCs expressed smooth muscle actin, but not vWF and cTnI (Figure 2H).

The migration ability of CCs was assessed. The percentage of gap closure of primary aCCs (28.7 ± 1.5%) was lower compared to young counterpart (44.5 ± 1.9%), measured at 8 h (*p* < 0.001) (Figure 2I). By clonogenically expanding the CCs, the migration of both young and aged CCs was improved by 13.7% and 16.9%, respectively. Nonetheless, the migration area repopulated by clonogenic aCCs were still lower than that of clonogenic yCCs (45.6 ± 1.3% vs. 58.3 ± 1.4%; *p* < 0.001). Representative images are shown in Figure 2J.

To further characterise the isolated CCs, both clonogenic yCCs and aCCs were tested for stemness markers using qPCR. These results demonstrated lower *Gata4*, *Sox2*, *Nkx2.5*, and *Tert* gene expression in clonogenic aCCs relative to clonogenic yCCs (Figure 2K). Clonogenic aCCs showed 2.4-fold, 14.3-fold, 1.9-fold lower gene expression in *Gata4*, *Sox2* and *Tert*, respectively (*p* < 0.01). However, clonogenic aCCs showed 3733-fold lower gene expression in *Nkx2.5*. Therefore, clonogenic CCs were used for subsequent experiment.

### 2.3. Co-culture with MSCs Can Improve the Growth Kinetics of aCCs 

In order to evaluate the effect of MSCs on aCCs, aCCs were tested either in direct MSC co-culture (aCC^MC^) or separated using transwell (aCC^TW^) and compared to aCC and yCC alone cultures. Percent of aCCs gap closure were assessed at 8 h. The results showed that 54.6 ± 4.4% of gaps were repopulated by aCC^MC^ in 8 h, which was greater than aCCs (44.7 ± 2.5%; *p* = 0.02) and aCC^TW^ (42.9 ± 2.6%; *p* < 0.01) (Figure 3B). To exclude the possibility that MSCs, but not aCCs, contributed to improved gap closure, the number migrated aCCs, identified via a fluorescent lipophilic cationic indocarbocyanine dye (DiI) staining, was counted. Not surprising majority MSCs, distinguished by anti-human nuclei staining, were found within repopulated gap area. However, the number of migrated aCC, beyond 100 µm from the start line increased by double (16 ± 2 cells) in MSC coculture as compared to aCC alone culture (8 ± 2 cells) (*p* = 0.02). (Figure 3C). 

To evaluate if MSC co-culture could improve proliferation of aCCs, aCCs were labelled with CFSE prior to MSC coculture, a fluorescein dye of which the fluorescence intensity is reduced to half following cell division. The proliferative index of aCC^MC^ and aCC^TW^ were 93.4 ± 3.3 and 94.7 ± 2.0, respectively, which were significantly higher than aCCs alone (81.3 ± 2.3, *p* < 0.01) (Figure 3D,E). 

### 2.4. aCC Cardiomyocyte Differentiation in MSC Co-Culture

To assess if MSCs could improve aCCs cardiomyocyte differentiation, aCCs were cultured in dexamethasone-based cardiac differentiation medium (CDM) for 21 days. Magnetic Particles Iron Oxide (MPIO) was used to label CCs for re-isolation using a magnet to facilitate post-experiment analysis (Figure 4A) [13]. Nevertheless, only *Tnni3* gene expression was upregulated aCC^MC^ after 21 days of differentiation (4.1 ± 0.2-fold vs. aCCs, *p* < 0.01) (Figure 4B). All human samples tested for the tested primers showed Ct value of > 35, affirming that the primers used were mouse-specific (Appendix A).

By using the same experimental design but without CC re-isolation (Figure 5A), cardiac differentiation of aCC^MC^ and aCC^TW^ directly were induced for 21 days. No differences in gene expressions were observed for *Myh7* and *Gata4* (Figure 5B). Nevertheless, *Myh6*, *Nkx2.5* and *Tnni3* gene expressions were significantly up-regulated in aCC^MC^ (4.4-fold, 6.5-fold, 4.1-fold, respectively) as compared to control aCCs (1.9-fold, 1.3-fold, 1.4-fold, respectively, *p* = 0.028 for *Myh6*; *p* < 0.001 for *Nkx2.5*; *p* = 0.01 for *Tnni3*) and aCC^TW^ (2.0-fold, 0.8-fold, 1.0-fold, 0.8-fold, respectively, *p* = 0.032 for *Myh6*; *p* < 0.001 for *Nkx2.5*; *p* < 0.01 for *Tnni3*). Representative images of cTnI staining were as shown in Figure 5C. 

### 2.5. Effects of MSCs Co-Culture on aCC Stemness, Telomere Length, and Telomerase Activity

Next, to evaluate the effects of MSC co-culture on aCC stemness gene expression, senescence assay, telomere length and telomerase activity, MPIO-labelled aCCs were first separated after being in MSC co-culture for five days prior to analysis. No significant difference was observed in cardiac markers *Gata4* and *Nkx2.5* and telomerase reverse transcriptase *Tert* (Figure 6A–C). However, there was a significant up-regulation in stemness marker *Sox2* expression in aCC^MC^ (1.8 ± 0.3-fold) as compared to aCCs (1.1 ± 0.1-fold) and aCC^TW^ (0.9 ± 0.2-fold) (*p* = 0.045) (Figure 6D). Marker of cellular senescence, *p16^INK4a^*, was found comparable in all groups, including yCCs and aCCs (Figure 6E). Co-culture with MSCs also did not down-regulate the *p16^INK4a^* gene expression in aCCs.

The number of senescent cells, identified by β-galactosidase stained cells, was found higher in aCCs as compared to yCCs, but no differences was found in aCCs with or without MSCs (Figure 6F). In addition, no difference was observed for telomere length and telomerase activity in aCC^MC^ and aCC^TW^, as compared to yCCs and aCCs alone cultures (Figure 6G,H). 

## 3. Discussion

Autologous cardiac cells were once a preferable source stem cells for use in therapy. However, it is known that the function of stem cells deteriorates as they age [14,15,16], casting doubts on whether autologous cell therapy is useful for elderly. Hence, in this study we examined and compared CCs isolated from both young and old mice. By adopting an isolation method commonly used in most studies, we sorted the c-kit expressing cardiac cells and depleted the lineage committed cells expressing CD5, CD11b, CD19, CD45R, Ly6G/C, and TER119. However, the method alone did not completely remove haematopoietic cells, evident by the presence of CD34 expressing cells in our primary culture (Appendix A). 

To yield highly proliferative CCs with minimal contamination from haemopoietic cells, we introduced a two-step clonogenic method by selecting colony-forming cells grown from primary CC culture seeded at a very low cell density (10000 cells/cm^2^), and clonally expanded them from single cells, a method truly attested to their self-renewal capability [11]. We also show that this method can remove haematopoietic, less proliferative cells from the population. Flow cytometric analysis showed minimal changes in surface marker expression in yCCs after the clonal expansion. These CCs were mostly CD166 positive but negative for CD90 and CD140. However, two distinct populations varying in CD90, CD140a, and CD166 expressions were observed in the primary aCCs. 

Clonal selection and expansion of primary aCCs revealed a predominant CD90^Pos^/CD140^Pos^/CD166^Neg^ phenotype, which was neither observed in primary nor clonogenic yCCs. Past studies have shown that CD90^Pos^/CD140^Pos^ cells were found in colony-forming unit cardiac mesenchymal-like stem cells derived from epicardium, [17] and ageing also caused an increase in CD90-expressing cells in cardiosphere-derived cells [18], the aggregate of cardiac progenitors from heart tissue explants which were sorted based on sphere-forming ability in response to growth factors [19,20]. Furthermore, CD166 or activated leukocyte cell adhesion molecule (ALCAM), an important marker which is expressed in cardiac progenitors involved in heart development [21,22] and morphogenesis in xenopus [23] and mice [21], was found present in yCCs but absent in the proliferative clonogenic aCC population, a similar expression pattern was also previously reported in senescence CCs [24]. Hence, it is possible that CD90^Pos^/CD140^Pos^/CD166^Neg^ aCCs resemble a more mesenchymal-like stem cell phenotype and its emergence in CC population is a result of ageing, [25,26,27] albeit require confirmation from in vivo study.

Nevertheless, as CCs represent a heterogeneous population, it is thus unclear whether the two populations (young and aged) represent two different cell types or share the same origin but evolve/change with ageing. We inclined towards the latter because the clonogenic aCCs may better represent the majority population in aged CCs after expansion due to its fast-growing characteristic, as long-term expansion is commonly needed to yield therapeutically relevant cell number. The primary aCCs revealed shorter telomeres as compared to yCCs, which suggest the correlation between telomere length and ageing [7]. Here, we showed that clonally expanded aCCs were mostly with preserved telomere length and telomerase activity. However, the preserved telomere length did not salvage the clonogenic aCCs from reduced growth kinetics and cardiac differentiation capability [28]. This may be attributed to its altered expressions of pluripotent gene (*Sox2),* cardiac progenitor markers (*Gata4*, *Nkx2.5*), and telomerase reverse transcriptase (*Tert*) which maintains telomere integrity [29,30,31].

Previous studies demonstrated that bone marrow MSCs show superior regenerative effects on damaged when administrated in combination with cardiac stem cells [8,32,33,34]. In view of the ageing effect on autologous cells, we chose a more biologically young Wharton’s Jelly MSCs, to coculture with aCCs. Exposing aged animals or cells to young microenvironment has been shown to reverse ageing phenotypes [35,36,37,38,39], but Wharton’s Jelly MSCs did not rejuvenate aCCs in co-culture. This may be attributed to insufficient MSC coculture time to observe an effect in aCCs which were chronically exposed to surrounding ageing stimuli in vivo [40], i.e., the senescence associated secretory phenotype (SASP) [41], a known cell senescent accelerator [42]. Nonetheless, our data shows that MSCs improved aCC migration and proliferation. Furthermore, greater cardiac and endothelial differentiation were also observed in aCC in direct MSC coculture, as compared to priming aCCs with MSCs in coculture prior to differentiation. Although the mechanism underlying this observation was not studied, but MSCs have been known to synergise various cells via juxtracrine and paracrine mechanisms [43].

MSC capability to secrete cardioprotective factors and communicate with other stem cells through paracrine signalling have been widely demonstrated [44,45]. Such paracrine effect has also been shown to offer therapeutic benefit in repair of damaged heart in vivo [46,47]. However, our data suggested that transwell co-culture of Wharton’s Jelly did not improve aCCs on any tested aspect. It is possible that secretomic profile of human Wharton’s Jelly MSCs is different from bone marrow derived MSCs (BMSCs), or the secreted factors fail to offer similar benefits in aged cells. Regardless, further in-depth analysis is required to better understand MSC secretome and its effects on aged cells.

The present study revealed that cardiac c-kit cells from elder subjects expressed different surface markers and growth kinetics, though with similar ageing profile with the cells from younger subjects. However, aCCs may be functionally compromised and required longer expansion time to acquire clinically relevant number. Co-culturing the aCCs with biologically young Wharton’s jelly MSCs in order to salvage the aCCs only minimally improved aCCs growth kinetics in vitro but did not drive rejuvenation. Hence, autologous CC therapy might not be ideal for elder patients, and a better alternative cell source needs to be sought.

In conclusion, clonogenic aCCs are mainly CD90^Pos^/CD140^Pos^/CD166^Neg^, which is absent in primary and clonogenic yCCs. The emergence of the population may be due to ageing, albeit requires further confirmation from in vivo study. The growth kinetic and cardiac differentiation capability of the clonogenic aCCs were expectedly inferior to yCCs and can be improved with Wharton’s Jelly MSCs in direct co-culture. Nonetheless, the effects observed in this study were minimal. Future studies will examine the feasibility and therapeutic efficacy of functionally augmented aged autologous cells comparing the young allogenic cells for treating elderly heart patients.

## 4. Materials and Methods

### 4.1. Isolation and Characterisation of Cardiac c-kit Cells (CCs) from Young and Aged Mice

The protocol for isolating endogenous CCs was adapted from Smits et al. (2009), with slight modification [11]. All procedures were performed according to the guidelines approved by the Institutional Animal Care and Use Committee (IACUC) of Universiti Sains Malaysia [USM/Animal Ethics Approval/2014/ (91) (547)] (Date of approval: 18th June 2014). Briefly, the whole heart from two mice was extracted from 1- and 18-month-old C57/BL6N mice immediately following carbon dioxide asphyxiation. Heart tissue was collected in ice-cold Dulbecco’s Modified Eagle Medium: Nutrient Mixture F-12 (DMEM/F12), supplemented with 20% foetal bovine serum, and 1 x Penicillin and Streptomycin (Gibco; Thermo Fisher Scientific, Chino, CA, USA). The collected heart was washed in cold-M buffer to remove residual blood by gently pressed with sterile forceps. Upon removal of non-heart tissues, the heart was minced into small pieces of about 1 mm^3^ and digested in 1 mg/mL Collagenase A (Roche Applied Science, Indianapolis, IN, USA) for 2 h at 37 °C in a water bath. Digested heart tissues were passed through a 40 µm cell strainer (BD Biosciences, Franklin Lakes, NJ, USA), ground using a syringe plunger and washed in cold M-buffer five times. Then, the cell filtrate was centrifuged at 300 *g* for 5 min at room temperature. Cell pellet was re-suspended in 5 mL of incubation medium, followed by sorting using EasySep Mouse c-kit positive selection cocktail (STEMCELL Technologies, Vancouver, Canada) and EasySep Mouse haematopoietic Progenitor Cell Isolation Cocktail (STEMCELL Technologies, Vancouver, Canada) according to the manufacturer’s protocols. Due to CC heterogeneity, positively selected CCs were plated as 1 cell per two wells on 96-well plate to establish clonogenically-expanded CCs. The selected cell colonies with at least 20 cells were cultured on 1.5% (*w*/*v*) gelatin-coated (Sigma-Aldrich, St. Louis, MO, USA) surface in cardiac cell complete growth medium (CGM) (see Appendix A) for subsequent clonogenic expansion. CCs were characterised by flow cytometry and immunocytochemistry using the antibodies as listed in Table 1.

Clonogenicity (%) = (Number of clones generated from single cell)/(Number of well with single cell at day 0) × 100%.(1)

### 4.2. Isolation and Characterisation of Wharton’s Jelly-Derived Mesenchymal Stem Cells

The human mesenchymal stem cells (MSCs) were a gift from Dr. Rajesh Ramasamy (Stem Cell & Immunity Research Group, Universiti Putra Malaysia). The isolation and characterisation were fully performed by their group according to Tong et al. (2011). Briefly, umbilical cord samples were collected from mother at full-term pregnancy with informed consent in accordance to ethical committee from Faculty of Medicine and Health Sciences, Universiti Putra Malaysia. The Wharton’s Jelly was collected after removal of blood vessels and minced into paste-like tissues. The tissues were incubated with 0.4% collagenase type II and 0.01% DNase at 37 °C for 30 min with gentle agitation. Equal volume of MSC growth medium, which was composed of Dulbecco’s Modified Eagle Medium: Low Glucose (DMEM/LG) supplemented with 1% (*v*/*v*) Penicillin/Streptomycin and 10% (*v*/*v*) Foetal Bovine Serum (FBS) was added to stop the enzymatic reaction. The digested tissues were then homogenised using handheld cell homogeniser for 5 min. This was followed by filtration through 40 µm cell strainers. The cells were then centrifuged and seeded onto T25 culture flask in MSC growth medium at 1 × 10^6^ cells/cm^2^.

### 4.3. CC-MSC Co-Culture

Co-cultures were performed with human MSCs and clonogenic CCs at a density of 1500 cells/cm^2^ (ratio of 1:1) through either mixed cultures or separated using 3 µm transwell in medium consisting of 50% CGM and 50% MGM for five days. Young and aged CCs served as controls. To distinguish CCs from MSCs in mixed cultures, CCs were pre-labelled with a cell tracker, such as MPIO (50 µg/mL) or DiI (1 µM) prior to co-cultures.

### 4.4. Magnetic Particles Iron Oxide (MPIO) Labelling

The CCs were labelled with 50 µg/mL of MPIO 24 h prior co-culture. MPIO endocytosed into cell cytoplasm could be observed under fluorescence microscope. The cells were washed thrice with Dulbecco’s Phosphate Buffer Solution (DPBS) to remove excessive extracellular MPIO. MPIO-labelled cells were then selected using a magnet and seeded for subsequent experiments.

### 4.5. DiI Labelling

Stock solution was prepared by diluting 1 mg of DiI (Thermo Fisher Scientific, Chino, CA, USA) in 500 µL DMSO to obtain concentration of 2 mg/mL in incubation medium (see Appendix A). Cells were incubated with DiI at final concentration of 1 µM for 5 min at 37 °C, followed by 15 min at 4 °C. Cells were washed twice with incubation medium. Final cell pellet was re-suspended in fresh medium and seeded accordingly.

### 4.6. Flow Cytometry

Cells were detached using Accutase (Gibco, Thermo Fisher Scientific, Chino, CA, USA) and centrifuged at 300 *g* for 5 min. The cells were then re-suspended in incubation medium and approximately 2 × 10^5^ cells were added into one 5 mL polypropylene tube. For direct labelling, cells were labelled with fluorescein isothiocyanate (FITC) or phycoerythrin (PE) conjugated antibodies at 4 °C in dark for 1 h. For indirect labelling, cells were first labelled with primary antibody for 1 h, followed by secondary antibody conjugated with Alexa Fluor 488 for another 1 h. The labelled cells were washed thrice with 500 µL DPBS at 300 *g* for 5 min prior to analysis with BD FACS Calibur flow cytometer (BD Biosciences, Franklin Lakes, NJ, USA).

### 4.7. Immunofluorescence Labelling

Cells were fixed with 4% (*w*/*v*) paraformaldehyde (PFA) (Acros, Branchburg, NJ, USA) for 20 min on ice, followed by being washed three times with PBS. For nuclear staining, cells were permeabilised with 0.1% (*v*/*v*) Triton X-100 (Sigma-Aldrich, St. Louis, MO, USA) for 10 min at RT. Upon three washes with 0.1% (*v*/*v*) Tween-20 in PBS (PBST), cells were blocked in either 10% (*v*/*v*) donkey or goat serum for 30 min at RT. Next, cells were incubated with respective antibodies at 4 °C overnight. Followed by three washes in PBST, cells were stained with secondary antibody conjugated with desired fluorochrome at RT for 1 h. After washing with PBST three times, the nuclei were counterstained with 1 µg/mL DAPI for 14 min at RT. After being washed with PBST three times, the cells were mounted with Vectashield mounting medium (Vector Laboratories, Burlingame, CA, USA). All images were viewed using Olympus IX71 fluorescence microscope (Olympus, Waltham, MA, USA).

### 4.8. CardioStem Sphere Formation and Trilineage Differentiation

To assess the function of the isolated CCs, CCs were characterised based on its capability to form cardiomyocytes, smooth muscle cells and vascular endothelial cells through the formation of CardioStem Spheres (CSps). Briefly, approximately 2 × 10^4^ CCs were plated onto a 100 mm bacteriological-grade petri dish in 5 mL leukemic inhibitory factor- (LIF) deprived CGM to generate CSps. The generated CSps were then plated onto laminin-coated (1 µg/mL) Lab-Tek 8 well-chamber slides and incubated in 5% CO_2_ incubator at 37^o^C for 7 days and the medium was changed every three days. Samples were collected and fixed with 4% PFA on ice at day 7 and stained for cardiac troponin I (cTnI) (Santa Cruz Biotechnology, Dallas, TX, USA), smooth muscle actinin (SMA) (Sigma-Aldrich, St. Louis, MO, USA), and von Willebrand factor (vWF) (Dako, Denmark) to detect successful differentiation.

### 4.9. Chemical-Induced CC Cardiac Differentiation

To assess for CC cardiac differentiation, CCs were plated at a density of 10000 cells/cm^2^ onto 1 µg/mL laminin-coated plates. The cells were allowed to grow to full confluency before treated with cardiac differentiation medium (see Appendix A) for 21 days. Medium was refreshed every two days. The differentiated myocytes were identified using qPCR and immunofluorescence labelling (cTnI).

### 4.10. RNA Extraction and cDNA Conversion

Cells were harvested and RNA was extracted using RNeasy Mini Kit (Qiagen, Hilden, Germany) according to manufacturer’s protocol. RNA was stored at −80 °C or used directly for cDNA conversion. To quantitate amount of RNA, 1 µL of sample was loaded onto Nanodrop (Thermo Fisher Scientific, Chino, CA, USA). One µg RNA was then reverse transcribed to cDNA by using the QuantiTect Reverse Transcriptase kit (Qiagen, Hilden, Germany) according to manufacturer’s protocol. cDNA was either use directly or stored at −80 °C.

### 4.11. DNA Extraction

Cells grown in a monolayer was detached from the culture flask by either trypsinisation or using cell scraper. DNA was extracted using DNA mini kit (Qiagen, Hilden, Germany) according to manufacturer protocols and stored at −80 °C.

### 4.12. Protein Extraction

A total of 1 × 10^5^ cells were lysed in 100 µL of CHAPS lysis buffer (see Appendix A) for 30 min on ice. The lysate was centrifuged at speed 16000× *g* for 20 min at 4 °C. Lysate was aliquot and frozen in liquid nitrogen and stored at −80 °C. Five µL were aliquoted separately and heat inactivated for 10 min at 85 °C to inhibit telomerase activity.

### 4.13. Quantitative Real-Time PCR

Quantitative real-time PCR (QPCR) was performed using the Quantinova SYBR Green Kit (Qiagen, Hilden, Germany), and the fluorescence intensity was measured by StepOne Plus (Applied Biosystem, Foster CA, USA) according to manufacturers’ protocols. Briefly, a reaction master mix was prepared by mixing 10 µL 2x QuantiNova SYBR Green PCR Master Mix, 2 µL of QN ROX reference dye, forward and reverse primer at final concentration of 0.7 µM each, and 50 ng of cDNA per one reaction. The cycling condition was i) PCR initial heat activation at 95 °C for 2 min, ii) 40 cycles of denaturation at 95 °C for 5 s and combined annealing/extension at 60 °C for 10 s, and iii) Melting curve analysis. The final Ct value was normalised to the housekeeping gene and fold changes were analysed using comparative the Ct value to a suitable control. List of primers was listed in Table 2.

### 4.14. Telomere Length Assessment

Average telomere length was measured from total genomic mouse DNA by using a real-time quantitative PCR method as previously described [48,49]. This assay is aimed to measure an average telomere length ratio by quantifying telomeric DNA with specially designed primer sequences, as shown in Table 3, and divided by the amount of a single-copy gene. Thirty-five ng of DNA sample were then used for the assay. For each standard curve, one reference DNA sample was diluted serially. The 36B4 gene served as the single-copy gene for normalisation. The cycling conditions were as shown in Table 4.

### 4.15. Real-Time Quantitative Telomeric Repeat Amplification Protocol (RQ-TRAP)

Cells were lysed in CHAPS lysis buffer and protein was used for TRAP-quantification. To generate a standard curve, protein lysates were prepared from 10000, 1000, 100, and 10 cells. One thousand cells protein lysate was used for the qPCR reaction. The average telomerase activity was performed according to previously published protocol [50] using specific primers and oligos as stated in Table 5. The cycling conditions were run according to Table 6. Telomerase activity was expressed relative to young control.

### 4.16. Proliferation Assessment of Aged CCs

Carboxyfluorescein (CFSE) was purchased from Gibco. Stock solution was prepared by diluting CFSE in 18 µL of DMSO to yield a stock concentration of 5 mM. One µL of CFSE stock was added into 1 mL cell suspension with cell number ≤ 1 × 10^6^ cells in incubation medium and incubated at RT for 20 min. Five times the volume of ice-cold medium with 5% serum was added to stop the reaction by incubating at RT for 5 min. Cells were then washed twice with incubation medium and re-suspended in either incubation or growth medium for further analysis or seeding, respectively. To verify successful labelling, labelled cells were analysed using flow cytometry. Cells were analysed for CFSE intensity on day three and analysed using MODFIT LT3.5 (Verify Software House, Inc, Topsham, ME, USA).

### 4.17. Senescence Assay

To stain for senescence cells, cells were washed once with DPBS, followed by fixing with 1 x Fixative Solution for 15 min according to manufacturer’s protocol (Cell Signalling Technology, Danvers, MA, USA). Next, the cells were incubated with β-galactosidase staining solution. The plate was sealed with parafilm to prevent evaporation and incubated at 37 °C for 16 h in a dry incubator. Finally, senescence cells were viewed and counted under a phase contrast microscope.

### 4.18. Viability Assay

Proliferation status of CCs were determined by using Presto Blue (Thermo Fisher Scientific, Chino, CA, USA). Briefly, CCs were seeded at 500 cells per well onto 96-well plates. Different cell numbers (315, 625, 1250, 2500, 5000, 10,000, and 20,000 cells) were seeded to generate a relative standard curve to quantitate the exact cell number following three days of culture. Ten percent Presto Blue in serum free medium were added into the culture and incubated for 1 h at 37 °C. The value of fluorescence intensity was measured at 544 nm/590 nm with the use of microplate reader (BMG Labtech, Germany).

### 4.19. Migration

Cells were seeded on iBIDI cell culture insert (iBIDI, Germany) for 24 h. The next day, thew cell culture insert was removed and washed once with DPBS. The removal of cell culture inserts allows cells to migrate into the gap created by the barrier. Images were captured at 10 x magnification at 0 h and 8 h. The area which recolonized by cells were analysed using image J.

Gap Closure Percentage (%) = (Gap Area (8 h-Baseline))/(Gap Area Baseline) × 100%.(2)

### 4.20. Data Analysis

All data were expressed as mean ± standard error of mean (SEM). All statistical analyses were performed using SPSS (IBM SPSS Statistics 22). The differences between groups were analysed using independent T-test and One-way ANOVA with Tukey post-hoc test and were considered significant when *p* < 0.05.

## Figures and Tables

**Figure 1 ijms-20-05519-f001:**
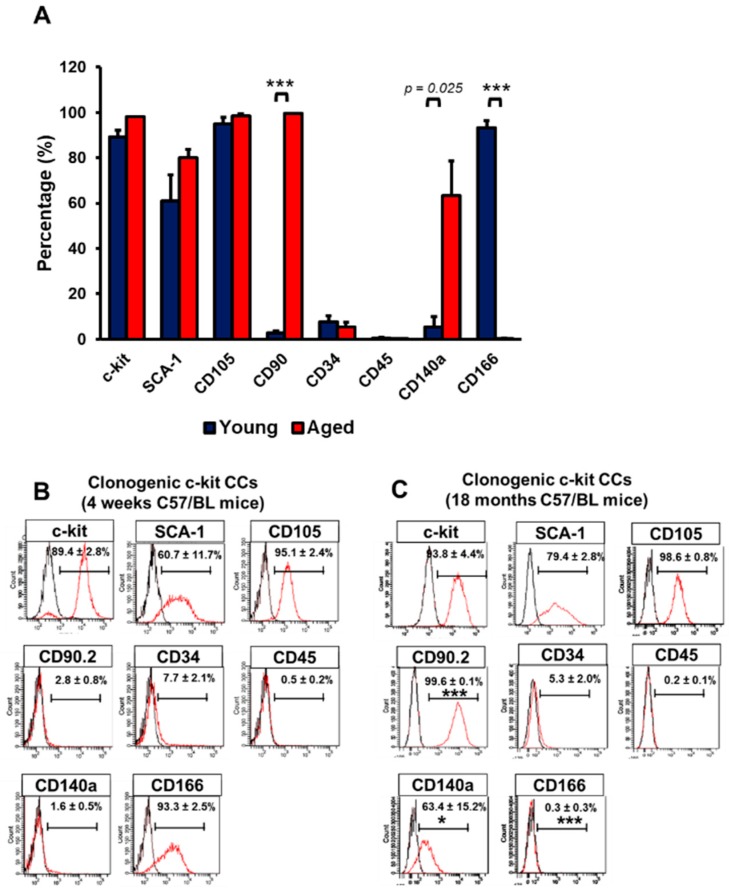
Clonogenic cardiac c-kit cells (CCs) can be derived from single cells. Clonogenic CCs were cultured from single cell for 14 days. (**A**) Graph showing surface marker expressions on c-kit-expressing CCs isolated from young (1-month) and aged (18-months) C57/BL6N mice. (**B**) Surface marker expression of clonogenic (**B**) young and (**C**) aged CCs. All data are mean ± SEM. * *p* < 0.05; *** *p* < 0.001 vs. Clonogenic yCCs (*n* = 3). [Abbreviation: CD34, hematopoietic marker; CD45, mast cells; CD90, thymocyte differentiation antigen; CD105, endoglin; c-kit, stem cell growth factor receptor; CD140a, PDGFRα; Sca-1, stem cell antigen; CD166, ALCAM].

**Figure 2 ijms-20-05519-f002:**
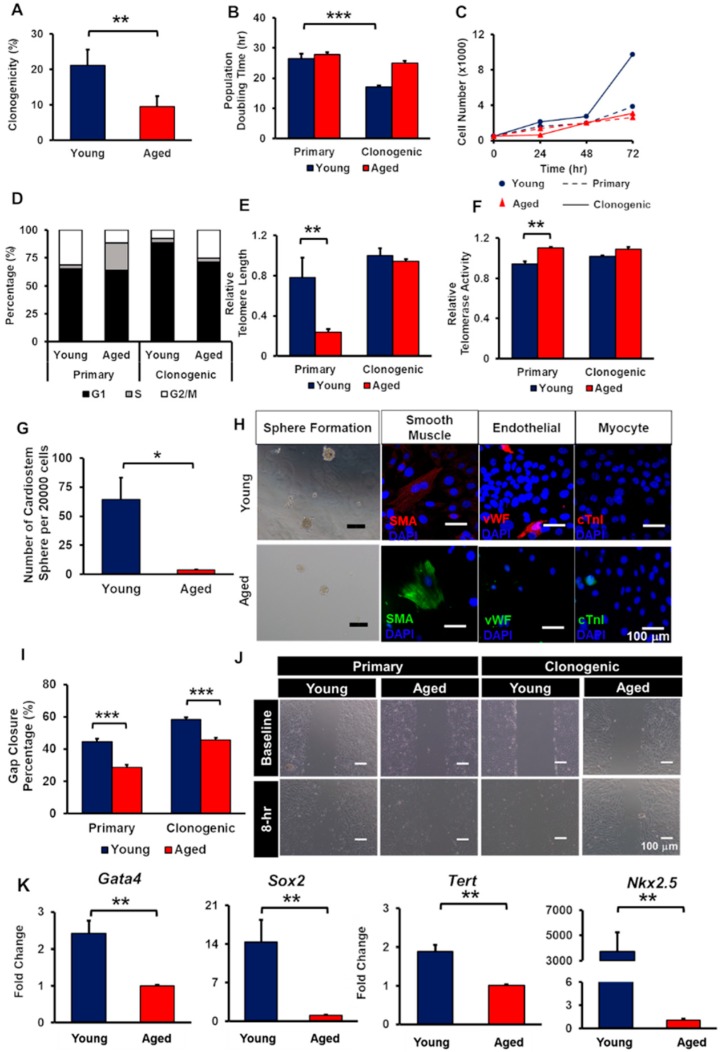
Clonogenically-expanded aged CCs (aCCs) show comparable telomere length to young CCs (yCCs) but functionally impaired compared to clonogenic yCCs. (**A**) Clonogenicity of yCCs and aCCs. (**B**) Population doubling time, (**C**) growth curve, (**D**) cell cycle, (**E**) relative telomere length, and (**F**) telomerase activity of primary isolated and clonogenic CCs. (**G**–**H**) CardioStem sphere formation and trilineage differentiation. (**I**) migration of both primary and clonogenic CCs. (**J**) Representative images of migrated cells after 8 h. (**K**) Stemness gene expressions. Scale bar = 100 μm. All data are mean ± SEM. * *p* < 0.05; ** *p* < 0.01; *** *p* < 0.001. (*n* = 3).

**Figure 3 ijms-20-05519-f003:**
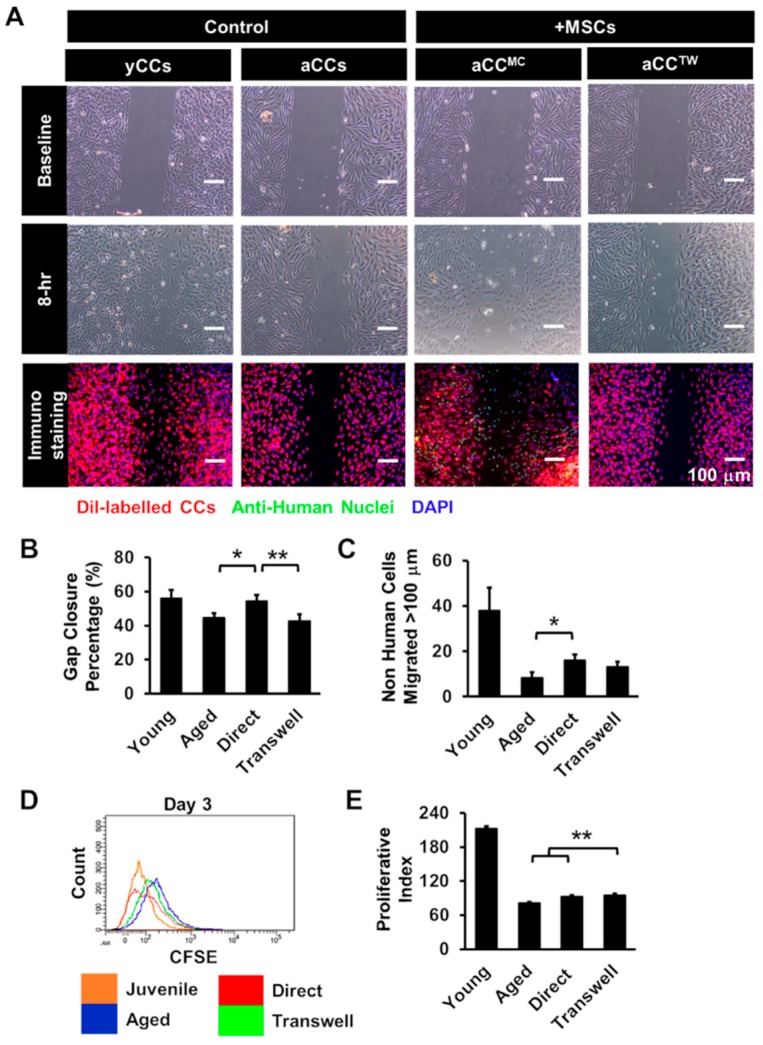
Co-culture with mesenchymal stem cells (MSCs) is capable of improving growth kinetics of aCCs. (**A**) Representative images of migrated cells after 8 h. CCs were labelled with DiI and MSCs were labelled with anti-human nuclei. Nuclei were stained in blue (DAPI). Scale bar = 100 μm. (**B**) Histogram showed gap area repopulated by CCs following co-culture. (**C**) Number of CCs moved more than 100 µm from the origin following co-culture. (**D**,**E**) Proliferative index of CFSE-labelled CCs following co-culture for three days. All data are mean ± SEM. * *p* < 0.05; ** *p* < 0.01. (*n* = 3).

**Figure 4 ijms-20-05519-f004:**
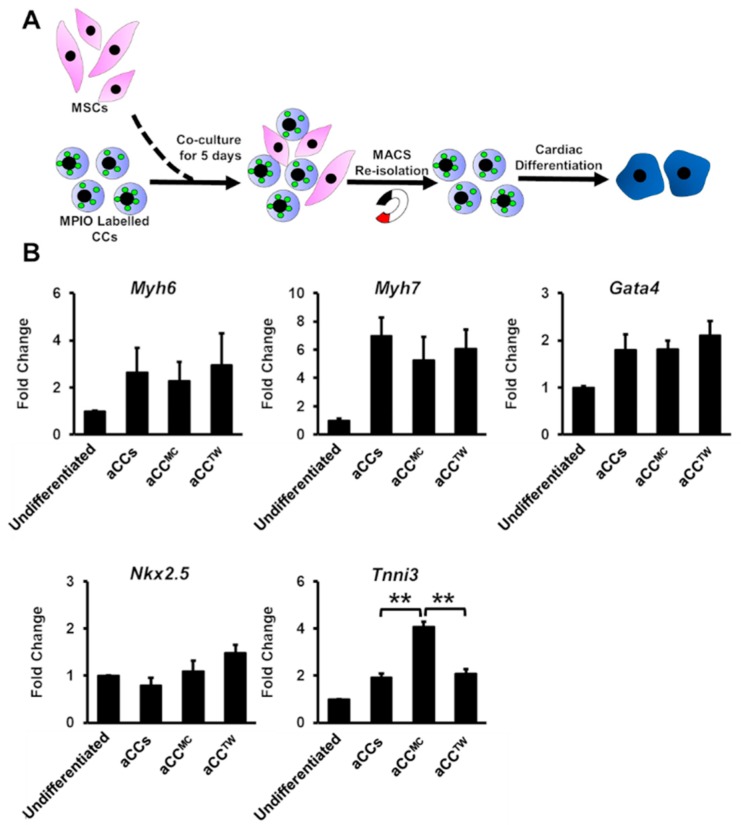
Co-culture with MSCs minimally improving the cardiomyocyte differentiation of aCCs. (**A**) Schematic diagram showing the cardiac differentiation of re-isolated CCs after co-culture. (**B**) Cardiac gene markers (*Myh6*, *Myh7*, *Gata4*, *Nkx2.5*, and *Tnni3*) were performed by qPCR. All data are mean ± SEM. ** *p* < 0.01. (*n* = 3).

**Figure 5 ijms-20-05519-f005:**
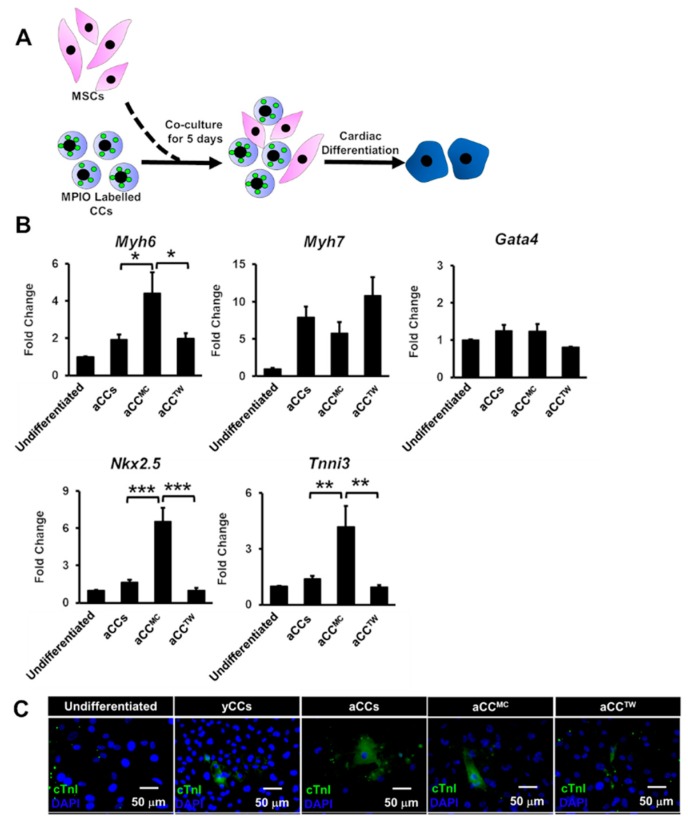
Co-culture with MSCs is capable of improving the cardiomyocyte differentiation of aCCs without CC reisolation. (**A**) Schematic diagram showing the cardiac differentiation of CCs without re-isolation after co-culture with MSCs. (**B**) Cardiac gene markers (*Myh6*, *Myh7*, *Gata4*, *Nkx2.5*, and *Tnni3*) were performed by qPCR. (**C**) Representative images of differentiated cells after 21 days. cTnI was stained in green. Nuclei were stained in blue (DAPI). Scale bar = 50 μm. All data are mean ± SEM. * *p* < 0.05; ** *p* < 0.01; *** *p* < 0.001. (*n* = 3).

**Figure 6 ijms-20-05519-f006:**
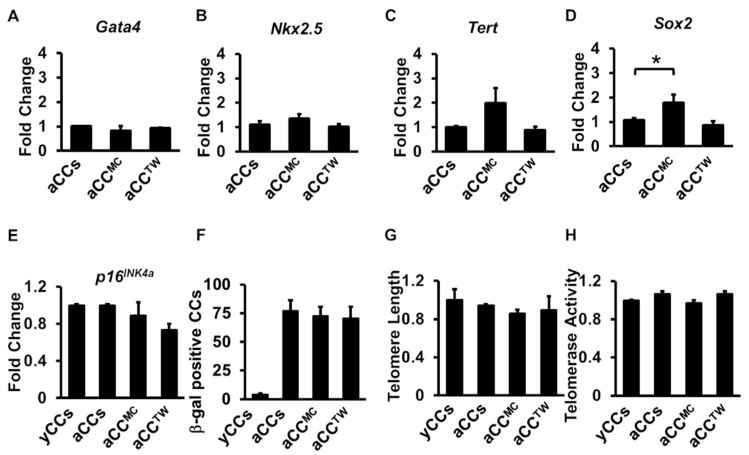
Co-culture with MSCs failed to rejuvenate aged CCs in five days of co-culture. (**A**–**D**) Stemness gene expressions (*Gata4, Sox2, Tert*, and *Nkx2.5*) as performed by qPCR. (**E**) *p16^INK4a^* gene expression of CCs (**F**) β-galactosidase assay for detection of senescent cells. (**G**) Relative telomere length and (**H**) relative telomerase activity of CCs following co-culture. All data are mean ± SEM. * *p* < 0.05. (*n* = 3).

**Table 1 ijms-20-05519-t001:** List of antibodies for CC characterisation and differentiation.

Antibody.	Dilution Factor	Application	Manufacturer
Rabbit Polyclonal Anti-c-kit Antibody (H-300)	1:50	ICC/FC	Santa Cruz Biotechnology, Germany (sc-5535)
FITC Rat Anti-mouse CD34 Antibody (Clone RAM34)	1:50	FC	BD Bioscience, USA (560238)
FITC Rat Anti-mouse CD90.2 Antibody (Clone 53-2.1,RUO)	1:50	FC	BD Bioscience, USA (553003)
FITC Rat Anti-mouse Sca-1 Antibody (Clone D7)	1:10	FC	Miltenyi Biotec, Germany (130-102-297)
PE Rat Anti-mouse CD140a Antibody (Clone APA5)	1:10	FC	Miltenyi Biotec, Germany (130-102-502)
PE Rat Anti-mouse CD166 Antibody (Clone eBioALC48)	1:50	FC	eBioscience, USA (12-1661-81)
PE Rat Anti-mouse CD105 Antibody (Clone MJ7/18)	1:10	FC	Miltenyi Biotec, Germany (130-102-548)
FITC Rat Anti-mouse CD45 Antibody (Clone 30F11)	1:10	FC	Miltenyi Biotec, Germany (130-102-491)
Rabbit Polyclonal Anti-GATA-4 Antibody (H-112)	1:50	ICC	Santa Cruz Biotechnology, Germany (sc-9053)
Rabbit Polyclonal Anti-NKX2.5 Antibody (H-114)	1:50	ICC	Santa Cruz Biotechnology, Germany (sc-14033)
Rabbit Polyclonal Anti-Ki67 Antibody (SP6)	1:50	ICC	Genetex, Germany (GTX16667)
Goat Polyclonal Anti-Tryptase Antibody (V-13)	1:50	ICC	Santa Cruz Biotechnology, Germany (sc-32473)
Goat Polyclonal Anti-Sox2 Antibody (Y-17)	1:50	ICC	Santa Cruz Biotechnology, Germany (sc-17320)
Rabbit Polyclonal Anti-OCT3/4 Antibody (H-134)	1:50	ICC	Santa Cruz Biotechnology, Germany (sc-9081)
Mouse Monoclonal Anti-Smooth Mucle Actinin (Clone 5C5)	1:400	ICC	Sigma Aldrich, USA (A2172)
Rabbit Polyclonal Anti-von Willebrand Factor Antibody	1:400	ICC	Dako, USA (A0082)
Rabbit Polyclonal anti-Cardiac Troponin I (H-170)	1:50	ICC	Santa Cruz Biotechnology, Germany (sc-15368)
Alexa Fluor 488 Donkey Anti-rabbit Antibody	1:500	ICC	Molecular Probes, CA
Alexa Fluor 488 Donkey Anti-goat Antibody	1:500	ICC	Molecular Probes, CA
Alexa Fluor 568 Donkey Anti-rabbit Antibody	1:500	ICC	Molecular Probes, CA
Alexa Fluor 568 Donkey Anti-goat Antibody	1:500	ICC	Molecular Probes, CA

Abbreviations: FITC = fluorescein isothiocyanate; PE = phycoerythrin; ICC = Immunocytochemistry; FC = Flow cytometry.

**Table 2 ijms-20-05519-t002:** Primer list used in this study.

Gene/Accession Number	Primer Sequence (5′-3′)
*Gata4* NM_008092.3	Forward: TCTCTGCATGTCCCATACCA Reverse: TGTGTGTGAAGGGGTGAAAA
*Nkx2.5* NM_008700.2	Forward: GCTACAAGTGCAAGCGACAG Reverse: GGGTAGGCGTTGTAGCCATA
*Sox2* NM_011443.3	Forward: GCGGAGTGGAAACTTTTGTCC Reverse: CGGGAAGCGTGTACTTATCCTT
*Tert* NM_009354.1	Forward: TGGGTCTCCCCTGTACCAAAT Reverse: GGCCTGTAACTAGCGGACACA
*Myh6* NM_010856.4	Forward: AAGGTGAAGGCCTACAAGCG Reverse: GGTCTGCTGGAGAGGTTATTCC
*Myh7* NM_080728.2	Forward: GCCAACACCAACCTGTCCAAGTTC Reverse: TGCAAAGGCTCCAGGTCTGAGGGC
*Tnni3* NM_000353.4	Forward: TCTGCCAACTACCGAGCCTAT Reverse: CTCTTCTGCCTCTCGTTCCAT
*p16^INK4a^* NM_009877.2	Forward: CGCAGGTTCTTGGTCACTGT Reverse: TGTTCACGAAAGCCAGAGCG
*Gapdh* NM_008084.2	Forward: ACCCAGAAGACTGTGGATGG Reverse: CACATTGGGGGTAGGAACAC

**Table 3 ijms-20-05519-t003:** Primer used in telomere length assessment.

Gene	Primer Sequence (5′-3′)
*Telomere*	Forward: CGGTTTGTTTGGGTTTGGGTTTGGGTTTGGGTTTGGGTT Reverse: GGCTTGCCTTACCCTTACCCTTACCCTTACCCTTACCCT
*36B4*	Forward: ACTGGTCTAGGACCCGAGAAG Reverse: TCAATGGTGCCTCTGGAGATT

**Table 4 ijms-20-05519-t004:** Cycling condition for telomere length assessment.

Stage	Time	Temperature
Holding Stage	2 min	95 °C
Cycling Stage (For telomere sequence)		
(i) Denaturation	15 s	95 °C
(ii) Annealing extension	1 min	56 °C
Cycling Stage (For 36B4 sequence)		
(i) Denaturation	15 s	95 °C
(ii) Annealing	20 s	52 °C
(iii) Extension	30 s	72 °C

**Table 5 ijms-20-05519-t005:** Primer used in telomerase activity assessment.

Gene	Primer Sequence (5′-3′)
*TS substrate*	AATCCGTCGAGCAGAGTT
*ACX (Anchored primers)*	GCGCGGCTTACCCTTACCCTTACCCTAACC

**Table 6 ijms-20-05519-t006:** Cycling condition for telomerase activity assessment.

Stage	Time	Temperature
Holding Stage	20 min	25 °C
Cycling Stage		
(i) Denaturation	30 s	95 °C
(ii) Annealing extension	90 s	60 °C

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
