# Peer review of "Human Wharton’s Jelly-Derived Mesenchymal Stem Cells Minimally Improve the Growth Kinetics and Cardiomyocyte Differentiation of Aged Murine Cardiac c-kit Cells in In Vitro without Rejuvenating Effect"

_ijms, 2019, doi:10.3390/ijms20225519_

Round 1

Reviewer 1 Report

Great article with results of study that characterised cardiac c-kit cells (CCs) from aged mice and about the effects of human Wharton’s Jelly-derived mesenchymal stem cells (MSCs) on the growth kinetics and cardiac differentiation of aged CCs in vitro. It was estimated that MSCs can improve the growth kinetics of aged CCs. Material, description of methods, statistical processing and results are quite convincing and credible.

But the article needs in some minor revision - it is necessary to describe how the results can be used in practice in the future? And what is the practical value of the results of this pilot study? 

Reviewer 2 Report

This is an interesting paper that addresses the issue of Wharton’s Jelly-derived mesenchymal stem cells (MSCs) action on growth and differentiation of cardiac c-kit cells (CCs) in mouse model of heart regeneration. The authors also ask if MSCs could rejuvenate ageing CCs. The manuscript is of interest for readers but it needs some minor corrections and update.

Title:

Title should be updated to:

„Human Wharton’s Jelly-Derived Mesenchymal Stem Cells Can Improve the Growth Kinetics and Cardiomyocyte Differentiation of Aged Murine Cardiac c-kit Cells in Vitro without Rejuvenating Effect”.

Correct “characterisation” to “characterization” in text.

Abstract:

Line 19. correct to “…and were directly co-cultured with MSCs…”

Line 51. correct to “…can be enhanced in the presence…”

Line 54. Remove the supernumerary space in front of “Here, we sought…”

Line 54. Remove the supernumerary space in front of “…growth kinetics and… ”

Line 56. correct to “…derived MSCs co-culture…”

Results:

Line 64. correct to “Characterization of the…primary c-kit cells was…”

Line 66. correct to “Wells with CC colony of more…”

Line 73. correct to “…vs. CD140a+ (63.4 + 15.2%)…”

Line 75. Add lost space in front of “Collectively, proliferative…”

Line 89. correct to “…at 1 cell/1 well for 2 weeks…”

Line 106. correct “…but with 20% lesser cells in G2 phase…”to “…but with almost 2.5-fold lower percentage of cells in G2 phase…”

Line 109. correct “…clonogenic aCCs showed 17% more cells resided in G2…” to “…clonogenic aCCs showed almost 3-fold higher percentage of cells resided in G2 phase…”

Figures:

Fig. 1. (A) Please change the ordinate range to show 100%.

Discussion:

Line 218. correct to “…colony-forming cells grown from primary…”

Line 220. correct to “…a method truly attested…”

Line 222. Remove the supernumerary space in front of “These CCs were…”

Line 247. correct to “This may be attributed to…”

Line 254-255. correct to “This may be attributed to…”

Line 261. correct to “…to synergize with…”

Line 263-264. correct to “Such paracrine effect has also been shown to offer therapeutic benefit in repair of damaged heart…”

Line 265. correct to “…transwell co-culture…”

Line 270. correct to “…the population may be due…”

Line 273. correct to “…MSCs in direct co-culture…”

Line 274. Remove the supernumerary space in front of “Future studies…”
